# Workspace Analysis and Path Planning of a Novel Robot Configuration with a 9-DOF Serial-Parallel Hybrid Manipulator (SPHM)

**Mahmoud Elsamanty** [1,2,*] , **Ehab M. Faidallah** [3], **Yehia H. Hossameldin** [4], **Saber Abd Rabbo** [1], **Shady A. Maged** [5] , **Hongbo Yang** [6,7] **and Kai Guo** [6,7,*]

1 Mechanical Department, Faculty of Engineering at Shoubra, Benha University, Cairo 11672, Egypt
2 Mechatronics and Robotics Department, School of Innovative Design Engineering, Egypt-Japan University of Science and Technology, Alexandria 21934, Egypt
3 Department of Mechatronics Engineering, The Higher Technological Institute, 10th of Ramadan City 44629, Egypt
4 Mechatronics Engineering Department, Future University, Cairo 11835, Egypt
5 Mechatronics Engineering Department, Ain Shams University, Cairo 11517, Egypt
6 Suzhou Institute of Biomedical Engineering and Technology, Chinese Academy of Sciences, Suzhou 215163, China
7 School of Biomedical Engineering (Suzhou), Division of Life Sciences and Medicine, University of Science and Technology of China, Hefei 230026, China
* Correspondence: mahmoud.alsamanty@feng.bu.edu.eg or mahmoud.elsamanty@ejust.edu.eg (M.E.); guok@sibet.ac.cn (K.G.)

**Abstract:** The development of serial or parallel manipulator robots is constantly increasing due to the need for faster productivity and higher accuracy. Therefore, researchers have turned to combining both mechanisms, sharing the advantage from serial to parallel or vice versa. This paper proposes a new configuration design for a serial-parallel hybrid manipulator (SPHM) using the industrial robotic KUKA Kr6 R900 and 3-DOF parallel spherical mechanism. The Kr6 R900 has six degrees of freedom (6-DOF) divided into three joints for translation (x, y, z) and another three joints for orientation (A, B, C) of the end-effector and the 3-DOF parallel spherical mechanism with three paired links. On the contrary, each limb of the parallel spherical mechanism consists of revolute–revolute–spherical joints (3-RRS). This mechanism allows translation movement along the Z-axis and orientation movements about the X- and Y- axes. The new hybrid will enrich the serial manipulator in movement flexibility and expand the workspace for serial and parallel manipulator robots. In addition, a complete conceptual design is presented in detail for the new robot configuration with a schematic and experimental setup. Then, a comprehensive mathematical model was derived and solved. The forward, inverse kinematics, and workspace analyses were derived using the graphical solution. Additionally, the new hybrid manipulator was tested for path planning. Moreover, an experimental setup was prepared to test the selected path. Finally, the new robot configuration can enlarge the workspace of both manipulators and the selected path matched to the experimental test.

**Keywords:** workspace analysis; path planning; hybrid manipulator; 3-RRS parallel robot; KUKA robot; 9-DOF; serial-parallel

## 1. Introduction

Undoubtedly, manipulator robots have many applications in many fields. Researchers are looking for hybrid manipulators to improve and develop these manipulators [1]. The combination of hybrid serial-parallel (HSP) manipulators has many advantages. Serial manipulators are distinguished in that they can provide a large workspace and simple control. In contrast, the parallel manipulator can provide high payload capacity and precision [2]. Furthermore, the hybrid manipulator is for the mechanism and can also be in

the control and algorithm [3]. In hybrid manipulators, there are many categories, such as hybrid serial-parallel (HSP) [4–7], hybrid parallel-serial (HPS), and hybrid parallel-parallel (HPP). Each combination has its features and properties. A mechanism with a seven-degree-of-freedom hybrid serial-parallel (HSP) arm of a humanoid robot manipulator has been proposed. The model comprises a 4-DOF serial robotic representing the shoulder and elbow joints connected with a 3-DOF parallel manipulator as a wrist joint [8]. Fusing the equilibrium and deformation coordination equations has also created a fully dynamic model.

Moreover, three different configuration designs for the hybrid robot have been implemented using SCARA and SCARA-Tau mechanisms to show their advantages and disadvantages [9]. To eliminate oscillations for the HSP manipulator, they presented a dynamic stabilizing strategy to create an equilibrium between the swaying force and the swaying moment consisting of five interconnected links and having 3-DOF in the joint space [10]. Furthermore, the cable-linkage serial-parallel palletizing robot (CSPR) has been presented as a study and a design focused on a series-parallel hybrid device powered by flexible wires to reduce inertia and improve dynamical responsiveness [11]. They also obtained the kinematic and dynamic models of the system. A novel proposed HSP design with 6-DOF [12]. The mechanism combines a 3-P parallel with 3-DOF and a 3-PSP with 3-DOF in serial form. This new layout will help regulate the attitude of the soaking cover in an immersion lithography system. A 6-DOF hybrid manipulator was composed of the 3-SPR (spherical prismatic revolute joints) connected with the 3-RPS (revolute prismatic spherical joints). It presents a solution for the end-effector terminal constraint and mobility problems with another configuration composed of the 3-RPS connected with the 3-SPR serial-parallel manipulators [13].

The workspace analysis and actuation distribution of a novel $n$(3-RRS) mechanism were developed. This hybrid manipulator is designed to compose a serial-parallel mechanism for gripping non-cooperative objects in space [14]. A new ideal hierarchical methodology is proposed for a serial-parallel hybrid kinematic machine with five degrees of freedom [15]. In addition, the kinematics and dynamics model for the new system was obtained. The suggested method simultaneously empowers the machine to execute significant kinematic, dynamic, and challenging operations. Furthermore, a 9-DOF worm robot with a hybrid serial-parallel mechanism is designed with two improved segments of a 3-RPS mechanism. The findings demonstrate that the robot can move in any direction for any given distance, and the theoretical analysis and the actual locomotion performance agree [16]. A new finger with a serial-parallel mechanism using twisted and coiled polymers (TCP) is implemented and designed for excellent performance [17]. Moreover, the kinematic and dynamic model was shown for the mechanical hand's increased power-to-weight ratio. The results show that comparing experimental and theoretical data verified the finger's mechanistic kinematic design. Moreover, to climb the transmission tower, a unique series-parallel hybrid robot with three degrees of freedom translation, parallel legs, and a body linkage is planned and constructed [18]. Furthermore, the design is equipped with an electromagnet system to climb onto the transmission tower. The results show that the hybrid robot can climb at various angles and pass obstacles, and the magnetic attraction ensures stable climbing.

At the same time, other researchers have focused on parallel-serial hybrid mechanisms as previously designed and analyzed [19]. Further, there is a new segmental manipulator with a parallel-serial hybrid mechanism for robotized deburring of significant jet engine elements. The manipulator consists of a parallel planar platform with 3-DOF and a serial mechanical arm with 3-DOF. Additionally, closed-form symbolic solutions are obtained for forward and inverse displacement analyses.

A study into a five-degree-of-freedom manipulator with a parallel-serial structure provided a solution for the direct and inverse kinematic issues and workspace analysis [20]. The proposed manipulator allows for the accomplishment of five special moves, three for translations and two for the rotations movement patterns. The findings of the kinematic

analysis revealed that the solutions to both the inverse and the forward position issues may be determined analytically. A new proposed manipulator was developed to work as a subretinal insertion device in conjunction with intraoperative optical coherence tomography (OCT) [21]. A combination of two parallel mechanisms is connected in series to compose a parallel-serial robot with control of the remote center motion (RCM) for the needle (end effector). According to the assessment findings, the single control loop is operated at 15 ms, and the RCM control precision is within 1 mm. Another 5-DOF parallel/serial manipulator was presented [22], and a complete velocity and singularity analysis was derived. This hybrid system is composed of a parallel and serial mechanism similar to a tripod personified by double carriages running on opposite paths. This manipulator can successfully deliver the accomplishment of five special moves, three translations (3T), and two rotations (2R) movement patterns (3T2R) to its end effector.

A novel kinematic and dynamic modeling method was obtained [23]. A new 8-DOF hybrid manipulator was introduced that acts similarly to a human arm [24,25]. Moreover, they obtained the inverse kinematics of the system. The simulation experiment results revealed that the suggested has 32 solutions based on the identical target position, orientation matrix, and the stated redundant input variables, while the accuracy of the proposed technique for dealing with the inverse displacement was tested. On the other hand, the optimal motion planning methods for redundant manipulators were discussed and proposed to provide solutions with a variety of constraints, both linearly and nonlinearly [26–28].

This work aimed to propose and implement a hybrid manipulator robot that combines serial and parallel mechanisms with nine degrees of freedom. The hybrid robot's new configuration comprises a 6-DOF KUKA Kr6 R900 serial manipulator and a 3-RRS parallel manipulator with 3-DOF. The structure mechanism of the new SPHM is described in Section 2. Further, the kinematic model is deduced and developed to identify the new hybrid manipulator end-effector position and orientation related to the robot joint angles, as described in Section 3. The remaining paper is organized as follows: Modeling and simulation are discussed in Section 4. Workspace analysis for the SPHM is obtained in Section 5. Path planning was clarified in Section 6. The experimental setup is presented in Section 7. Finally, Section 8 presents the new conclusions and outcomes.

## 2. Structure Design of the Hybrid Manipulator

The new hybrid manipulator structure comprises two different types of manipulator robots. The first part is KUKA kr6 R900 (serial manipulator), while the second is a parallel manipulator with 3-RRS (three joints with revolute–revolute–spherical). This new configuration will be able to compose a neoteric design with nine degrees of freedom, as shown in Figure 1. In addition, Table 1 shows the hybrid manipulator joint range.

**Table 1.** Hybrid manipulator joint range.

| Joint No. | Value ° |
|---|---|
| Joint 1 | $\pm 170°$ |
| Joint 2 | $-190°/45°$ |
| Joint 3 | $-120°/156°$ |
| Joint 4 | $\pm 185°$ |
| Joint 5 | $\pm 120°$ |
| Joint 6 | $\pm 350°$ |
| Joint 7 | $0°/120°$ |
| Joint 8 | $0°/120°$ |
| Joint 9 | $0°/120°$ |

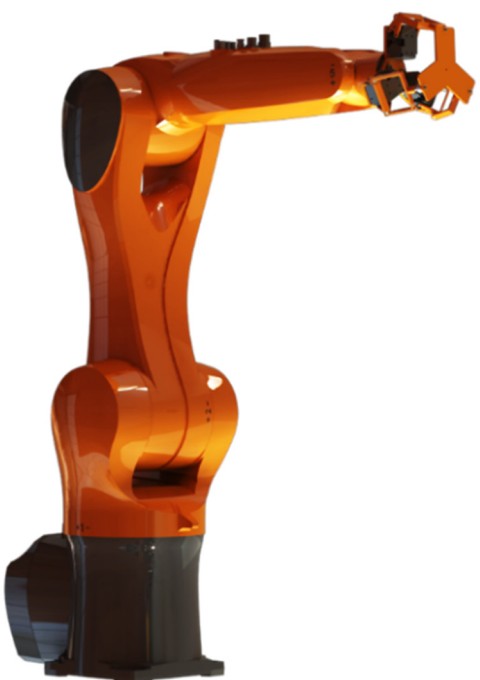

**Figure 1.** CAD Design for The New SPHM.

## 2.1. KUKA KR6 R900 Manipulator

The serial KUKA KR6 R900 manipulator has six degrees of freedom with six revolute joints, as shown in Figure 2a. The normal payload is 3 kg, while the ultimate payload is 6 kg with an end effector maximum reach of 901.5 mm, as shown in Figure 2b [29]. In addition, the main robot specifications are 52 kg for the total weight and the KRC4 compact controller. The most comprehensive application of the robot is assembly, pick and place, material handling, and packing.

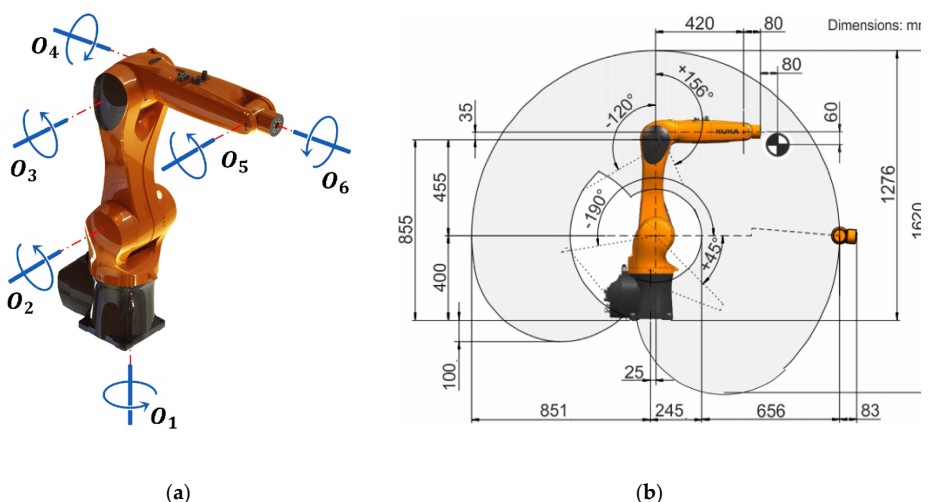

(**a**) (**b**)

**Figure 2.** KUKA KR6 R900 (**a**) Arbitrary posture with joints' orientation movement; (**b**) dimensions of KUKA KR6 R900.

## 2.2. 3-RRS Parallel Manipulator

On the other hand, a parallel manipulator (3-RRS) was implemented for the new hybrid manipulator, and the CAD model is presented in Figure 3. Roll ($\psi_x$), pitch ($\psi_y$), and heave ($O_7$) are the three degrees of freedom for the manipulator, which represents the position of the moving platform as $x = \{\psi_x, \psi_y, O_7\}^T$. The model comprises three limbs, and each one has an active link with length ($l_1$) and a passive link with length ($l_2$).

Moreover, it contains three joints with a revolute active joint at point ($A_i$), a revolute passive joint at the point ($C_i$), and a spherical joint at the joint ($B_i$). The active joint variable factors are grouped as $\theta = \{\theta_7, \theta_8, \theta_9\}^T$. While the passive joint variable factors are grouped as $\varnothing = \{\varnothing_1, \varnothing_2, \varnothing_3\}^T$. Therefore, the 3-RRS parallel robot contains a triangular base with center $O_6$, as shown in Figure 4a, and a triangular moving platform with center $O_7$; thus, $\beta_{a2} = \beta_{b2} = 120°$, $\beta_{a3} = \beta_{b3} = 240°$, as shown in Figure 4b,c. In addition, there are two coordinate systems $A(x, y, z)$ for the base, and $B(u, v, w)$ for the moving platform, which aid in the analysis. The $x$-axis was chosen in the direction of $\overline{O_6 A_1}$ and the $u$-axis was chosen in the direction of $\overline{O_7 B_1}$. In addition, three servo motors fixed at the base (one for each limb) were used to move ($\psi_x, \psi_y, O_7$) of the moving platform.

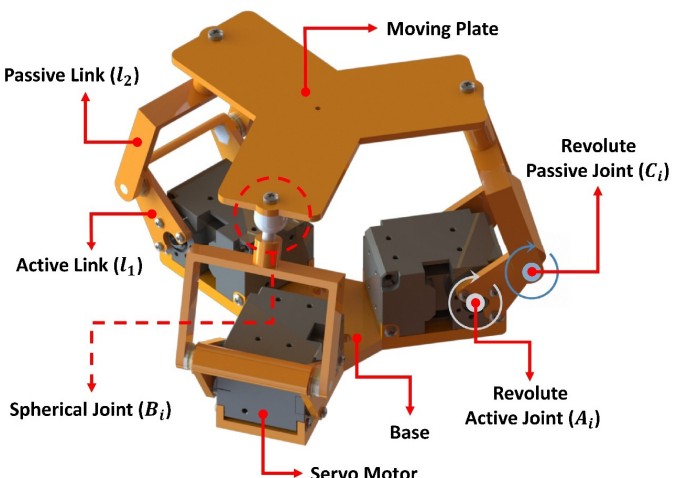

**Figure 3.** 3-RRS Manipulator CAD Model.

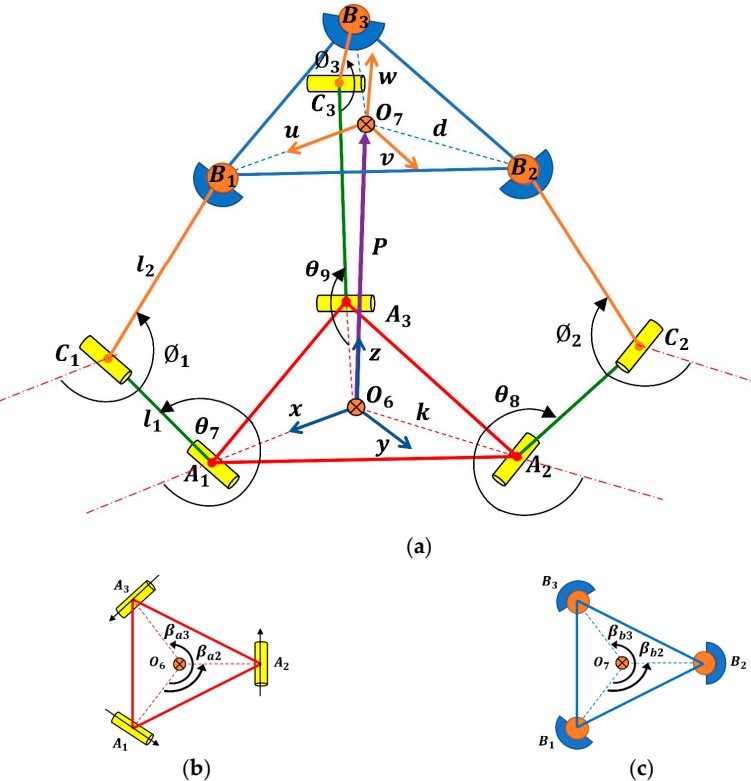

**Figure 4.** 3-RRS manipulator kinematic diagram. (**a**) Complete view; (**b**) fixed base (top view); (**c**) moving platform (top view).

## 3. Kinematic Analysis

Forward kinematics indicate that the input $\theta_i$ may determine the end-effector's final position and the orientation of the end effector. In this scenario, the novel hybrid manipulator with 9-DOF ($\theta_1 \ldots \theta_9$) is characterized as having a serial manipulator with six joint angles and a parallel manipulator with three joint angles. Where $\theta_1, \theta_2, \theta_3, \theta_4, \theta_5, \theta_6$ link $O_0O_1, O_1O_2, O_2O_3, O_3O_4, O_4O_5, O_5O_6$, respectively, $\theta_7, \theta_8$, and $\theta_9$ link $O_6O_7$, as illustrated in Figure 5.

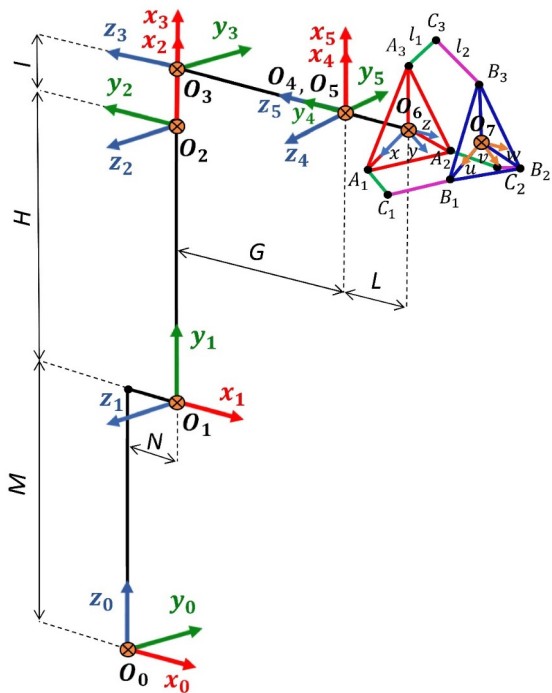

**Figure 5.** Kinematic Diagram for the New Serial-Parallel Manipulator.

The new forward kinematics of the hybrid manipulator can be solved by (1). This describes the transformation matrix between the frame $O_0$ and frame $O_7$.

$$T_7^0 = T_1^0 \, T_2^1 \, T_3^2 \, T_4^3 \, T_5^4 \, T_6^5 \, T_7^6 \tag{1}$$

Therefore, the position and orientation transformation matrix between $O_0$ and $O_6$ can be determined using the Denavit–Hartenberg (D-H) parameter and can be obtained as in Table 2.

**Table 2.** KUKA KR6 R900 (Serial Manipulator) D-H Parameters.

| Link $i$ | Link Twist $\alpha_i$ (Degree) | Link Length $a_i$ (mm) | Joint Offset $d_i$ (mm) | Joint Angle $\theta_i$ (Degree) |
|---|---|---|---|---|
| 0-1 | 90 | -N | M | $\theta_1$ |
| 1-2 | 0 | -H | 0 | $\theta_2 + 90$ |
| 2-3 | −90 | -I | 0 | $\theta_3$ |
| 3-4 | 90 | 0 | -G | $\theta_4$ |
| 4-5 | −90 | 0 | 0 | $\theta_5$ |
| 5-6 | 180 | 0 | -L | $\theta_6$ |

Therefore, $T_1^0$, $T_2^1$, $T_3^2$, $T_4^3$, $T_5^4$, and $T_6^5$ can be obtained by (2)–(8), where $C$ and $S$ represent *Cos* and *Sine*.

$$T_1^0 = \begin{bmatrix} C\theta_1 & 0 & S\theta_1 & -N.C\theta_1 \\ S\theta_1 & 0 & -C\theta_1 & -N.S\theta_1 \\ 0 & 1 & 0 & M \\ 0 & 0 & 0 & 1 \end{bmatrix} \tag{2}$$

$$T_2^1 = \begin{bmatrix} -S\theta_2 & -S\theta_2 & 0 & H.S\theta_2 \\ C\theta_2 & C\theta_2 & 0 & -H.C\theta_2 \\ 0 & 0 & 1 & 0 \\ 0 & 0 & 0 & 1 \end{bmatrix} \tag{3}$$

$$T_3^2 = \begin{bmatrix} C\theta_3 & 0 & -S\theta_3 & -I.C\theta_3 \\ S\theta_3 & 0 & C\theta_3 & -I.S\theta_3 \\ 0 & -1 & 0 & 0 \\ 0 & 0 & 0 & 1 \end{bmatrix} \tag{4}$$

$$T_4^3 = \begin{bmatrix} C\theta_4 & 0 & S\theta_4 & 0 \\ S\theta_4 & 0 & -C\theta_4 & 0 \\ 0 & 1 & 0 & -G \\ 0 & 0 & 0 & 1 \end{bmatrix} \tag{5}$$

$$T_5^4 = \begin{bmatrix} C\theta_5 & 0 & -S\theta_5 & 0 \\ S\theta_5 & 0 & C\theta_5 & 0 \\ 0 & -1 & 0 & 0 \\ 0 & 0 & 0 & 1 \end{bmatrix} \tag{6}$$

$$T_6^5 = \begin{bmatrix} C\theta_6 & S\theta_6 & 0 & 0 \\ S\theta_6 & -C\theta_6 & 0 & 0 \\ 0 & 0 & -1 & -L \\ 0 & 0 & 0 & 1 \end{bmatrix} \tag{7}$$

While (8) shows the transformation matrix between $O_6$ and $O_7$, where $u_x$, $u_y$, $u_z$, $v_x$, $v_y$, $v_z$, $w_x$, $w_y$, and $w_z$ are for orientation and $O_{7,x}$, $O_{7,y}$, and $O_{7,z}$ are for the position:

$$T_7^6 = \begin{bmatrix} u_x & v_x & w_x & O_{7,x} \\ u_y & v_y & w_y & O_{7,y} \\ u_z & v_z & w_z & O_{7,z} \\ 0 & 0 & 0 & 1 \end{bmatrix} \tag{8}$$

The transformation between the moving platform and the base can be described by a position vector: $P = \overline{O_6 O_7}$ and a $3 \times 3$ rotation matrix $R_B^A$, which describes the orientation of the moving platform with coordinate system B, with respect to the base with coordinate system A, as in Equation (9).

$$R_B^A = \begin{bmatrix} u_x & v_x & w_x \\ u_y & v_y & w_y \\ u_z & v_z & w_z \end{bmatrix} \tag{9}$$

To verify the rotation matrix $R_B^A$, its elements must satisfy the following orthogonal conditions (10)–(15) [30]:

$$u_x^2 + u_y^2 + u_z^2 = 1 \tag{10}$$

$$v_x^2 + v_y^2 + v_z^2 = 1 \tag{11}$$

$$w_x^2 + w_y^2 + w_z^2 = 1 \tag{12}$$

$$u_x v_x + u_y v_y + u_z w_z = 0 \tag{13}$$

$$u_x v_x + u_y v_y + u_z w_z = 0 \tag{14}$$

$$v_x w_x + v_y w_y + v_z w_z = 0 \tag{15}$$

where $a_i$ and $b_i$ are the position vectors of points $A_i$ and $B_i$ in the coordinate systems $A$ and $B$, respectively. Then, the coordinates of $A_i$ and $B_i$ are given by (16)–(21):

$$a_1 = [k,\, 0, 0]^T \tag{16}$$

$$a_2 = \left[ -\frac{1}{2}k,\, \frac{\sqrt{3}}{2}k, 0 \right]^T \tag{17}$$

$$a_3 = \left[ -\frac{1}{2}k,\, -\frac{\sqrt{3}}{2}k, 0 \right]^T \tag{18}$$

$$b_1 = [d,\, 0, 0]^T \tag{19}$$

$$b_2 = \left[ -\frac{1}{2}d,\, \frac{\sqrt{3}}{2}d, 0 \right]^T \tag{20}$$

$$b_3 = \left[ -\frac{1}{2}d,\, -\frac{\sqrt{3}}{2}d, 0 \right]^T \tag{21}$$

Equation (22) indicates the position vector of $B_i$ with respect to the base coordinate system.

$$B_i = p + R_B^A\, b_i \tag{22}$$

By substituting (9), (19), (20), and (21) into (22), the position vector $B_i$ can be obtained as (23)–(25):

$$\begin{bmatrix} B_{1,x} \\ B_{1,y} \\ B_{1,z} \end{bmatrix} = \begin{bmatrix} O_{7,x} \\ O_{7,y} \\ O_{7,z} \end{bmatrix} + R_B^A \begin{bmatrix} d \\ 0 \\ 0 \end{bmatrix} = \begin{bmatrix} O_{7,x} + du_x \\ O_{7,y} + du_y \\ O_{7,z} + du_z \end{bmatrix} \tag{23}$$

$$\begin{bmatrix} B_{2,x} \\ B_{2,y} \\ B_{2,z} \end{bmatrix} = \begin{bmatrix} O_{7,x} \\ O_{7,y} \\ O_{7,z} \end{bmatrix} + R_B^A \begin{bmatrix} -\frac{1}{2}d \\ \frac{\sqrt{3}}{2}d \\ 0 \end{bmatrix} = \begin{bmatrix} O_{7,x} - \frac{1}{2}du_x + \frac{\sqrt{3}}{2}dv_x \\ O_{7,y} - \frac{1}{2}du_y + \frac{\sqrt{3}}{2}dv_y \\ O_{7,z} - \frac{1}{2}du_z + \frac{\sqrt{3}}{2}dv_z \end{bmatrix} \tag{24}$$

$$\begin{bmatrix} B_{3,x} \\ B_{3,y} \\ B_{3,z} \end{bmatrix} = \begin{bmatrix} O_{7,x} \\ O_{7,y} \\ O_{7,z} \end{bmatrix} + R_B^A \begin{bmatrix} -\frac{1}{2}d \\ -\frac{\sqrt{3}}{2}d \\ 0 \end{bmatrix} = \begin{bmatrix} O_{7,x} - \frac{1}{2}du_x - \frac{\sqrt{3}}{2}dv_x \\ O_{7,y} - \frac{1}{2}du_y - \frac{\sqrt{3}}{2}dv_y \\ O_{7,z} - \frac{1}{2}du_z - \frac{\sqrt{3}}{2}dv_z \end{bmatrix} \tag{25}$$

There is a constraint for the motion of the limbs founded by each revolute joint $A_1$, $A_2$, and $A_3$. In Equation (26), $B_1$ is constrained on the $xz$ plane, and in Equation (27), $B_2$ is constrained on the $y = \tan(120°)x$ plane, while Equation (28) $B_3$ is constrained on the $y = \tan(240°)x$.

$$B_{1,y} = 0, \text{ for } A_1 \tag{26}$$

$$B_{2,y} = -\sqrt{3}B_{2,x}, \text{ for } A_2 \tag{27}$$

$$B_{3,y} = \sqrt{3}B_{3,x}, \text{ for } A_3 \tag{28}$$

By substituting the y-term of the position vector $B_i$ from (23)–(25) into (26)–(28), it obtains:

$$O_{7,y} + du_y = 0 \tag{29}$$

$$O_{7,y} - \frac{du_y}{2} + \frac{\sqrt{3}dv_y}{2} = -\sqrt{3}\,O_{7,x} + \frac{\sqrt{3}du_x}{2} - \frac{3dv_x}{2} \tag{30}$$

$$O_{7,y} - \frac{du_y}{2} - \frac{\sqrt{3}dv_y}{2} = \sqrt{3}\,O_{7,x} - \frac{\sqrt{3}du_x}{2} - \frac{3dv_x}{2} \tag{31}$$

By subtracting the sum of (30) and (31) from (29) multiplied by 2, we get

$$v_x = u_y \tag{32}$$

Furthermore, by subtracting (31) from (30).

$$O_{7,x} = \frac{d(u_x - v_y)}{2} \tag{33}$$

Therefore, (29) and (33) represent the constraint for the position of the moving platform, while (32) represents the constraint of the orientation of the moving platform. To compute forward kinematics, $\theta_7$, $\theta_8$, and $\theta_9$ are given, yet it needs to find $O_{7,z}$, $\psi_x$ and $\psi_y$. It can be solved by the position vector of the point $B_i$ with respect to the base coordinate system, which could be written as (34)–(36)

$$B_1 = \vec{O_6 B_1} = \begin{bmatrix} k + l_1 C\theta_7 + l_2 C\varnothing_1 \\ 0 \\ -l_1 S\theta_7 - l_2 S\varnothing_1 \end{bmatrix} \tag{34}$$

$$B_2 = \vec{O_6 B_2} = \begin{bmatrix} -\frac{1}{2}(k + l_1 C\theta_8 + l_2 C\varnothing_2) \\ \frac{\sqrt{3}}{2}(k + l_1 C\theta_8 + l_2 C\varnothing_2) \\ -l_1 S\theta_8 - l_2 S\varnothing_2 \end{bmatrix} \tag{35}$$

$$B_3 = \vec{O_6 B_3} = \begin{bmatrix} -\frac{1}{2}(k + l_1 C\theta_9 + l_2 C\varnothing_3) \\ -\frac{\sqrt{3}}{2}(k + l_1 C\theta_9 + l_2 C\varnothing_3) \\ -l_1 S\theta_9 - l_2 S\varnothing_3 \end{bmatrix} \tag{36}$$

While the moving platform has an equilateral triangle shape, the distance between every two points of $B_i$ is equal, and by using the cosine theorem $c^2 = a^2 + b^2 - 2ab cos(c)$, then, $|B_1 B_2|^2 = |B_2 B_3|^2 = |B_3 B_1|^2 = 3d^2$ [31].

$$\frac{1}{4}(3k + 2l_1 C\theta_7 + 2l_2 C\varnothing_1 + l_1 C\theta_8 + l_2 C\varnothing_2)^2 + \frac{3}{4}(k + l_1 C\theta_8 + l_2 C\varnothing_2)^2$$
$$+ (l_1 S\theta_7 + l_2 S\varnothing_1 - l_1 S\theta_8 - l_2 S\varnothing_2)^2 = 3d^2 \tag{37}$$

$$\frac{1}{4}(l_1 C\theta_8 + l_2 C\varnothing_2 - l_1 C\theta_9 - l_2 C\varnothing_3)^2 + (l_1 S\theta_8 + l_2 S\varnothing_2 - l_1 S\theta_9 - l_2 S\varnothing_3)^2$$
$$+ \frac{3}{4}(2k + l_1(C\theta_8 + C\theta_9) + l_2(C\varnothing_2 + C\varnothing_3))^2 = 3d^2 \tag{38}$$

$$\frac{1}{4}(3k + 2l_1 C\theta_7 + 2l_2 C\varnothing_1 + l_1 C\theta_9 + l_2 C\varnothing_3)^2 + \frac{3}{4}(k + l_1 C\theta_9 + l_2 C\varnothing_3)^2$$
$$+ (l_1 S\theta_7 + l_2 S\varnothing_1 - l_1 S\theta_9 - l_2 S\varnothing_3)^2 = 3d^2 \tag{39}$$

Rewrite (37)–(39) in terms of passive joint angle $\varnothing_i$:

$$e_{70} + e_{71} C\varnothing_1 + e_{72} C\varnothing_2 + e_{73} C\varnothing_1 C\varnothing_2 + e_{74} S\varnothing_1 + e_{75} S\varnothing_2 + e_{76} S\varnothing_1 S\varnothing_2 = 0 \tag{40}$$

$$e_{80} + e_{81} C\varnothing_2 + e_{82} C\varnothing_3 + e_{83} C\varnothing_2 C\varnothing_3 + e_{84} S\varnothing_2 + e_{85} S\varnothing_3 + e_{86} S\varnothing_2 S\varnothing_3 = 0 \tag{41}$$

$$e_{90} + e_{91} C\varnothing_3 + e_{92} C\varnothing_1 + e_{93} C\varnothing_3 C\varnothing_1 + e_{94} S\varnothing_3 + e_{95} S\varnothing_1 + e_{96} S\varnothing_3 S\varnothing_1 = 0 \tag{42}$$

where $i = 7, 8, 9$, and $i + 7 = 7$ *for* $i = 9$.

$$e_{i0} = 3k^2 + 2l_1^2 + 2l_2^2 - 3d^2 + l_1 C\theta_i(3k + l_1 C\theta_{i+1}) + 3kl_1 C\theta_{i+1} - 2l_1^2 S\theta_i S\theta_{i+1} \tag{43}$$

$$e_{i1} = l_2(3k + 2l_1 C\theta_i + l_1 C\theta_{i+1}) \tag{44}$$

$$e_{i2} = l_2(3k + l_1 C\theta_i + 2l_1 C\theta_{i+1}) \tag{45}$$

$$e_{i3} = l_2^2 \tag{46}$$

$$e_{i4} = -e_{i5} = 2l_1 l_2(S\theta_i - S\theta_{i+1}) \tag{47}$$

$$e_{i6} = -2l_2^2 \tag{48}$$

By substituting the formula of trigonometric identities formula: $S\varnothing_i = \frac{2t_{i}}{1+t_i^2}$ and $C\varnothing_i = \frac{1-t_i^2}{1+t_i^2}$ where $t_i = tan\frac{\varnothing_i}{2}$ into Equations (40) and (41), before rearranging them in terms of $\varnothing_2$ :

$$r_{70} + r_{71}S\varnothing_2 + r_{72}C\varnothing_2 = 0 \tag{49}$$

$$r_{80} + r_{81}S\varnothing_2 + r_{82}C\varnothing_2 = 0 \tag{50}$$

where

$$r_{70} = e_{70} + e_{71} + 2e_{74}t_1 + (e_{70} - e_{71})t_1^2 \tag{51}$$

$$r_{71} = e_{75} + 2e_{76}t_1 + e_{75}t_1^2 \tag{52}$$

$$r_{72} = e_{72} + e_{73} + (e_{72} - e_{73})t_1^2 \tag{53}$$

$$r_{80} = e_{80} + e_{82} + 2e_{85}t_3 + (e_{80} - e_{82})t_3^2 \tag{54}$$

$$r_{81} = e_{84} + 2e_{86}t_3 + e_{84}t_3^2 \tag{55}$$

$$r_{82} = e_{81} + e_{83} + (e_{81} - e_{83})t_3^2 \tag{56}$$

$C\varnothing_2$ and $S\varnothing_2$ can be obtained by using Equations (49) and (50):

$$C\varnothing_2 = \frac{r_{70}r_{81} - r_{71}r_{80}}{r_{71}r_{82} - r_{72}r_{81}} \tag{57}$$

$$S\varnothing_2 = \frac{r_{72}r_{80} - r_{70}r_{82}}{r_{71}r_{82} - r_{72}r_{81}} \tag{58}$$

Then,

$$\varnothing_2 = atan2(C\varnothing_2, S\varnothing_2) \tag{59}$$

by substituting Equations (57) and (58) into $C^2\varnothing_2 + S^2\varnothing_2 = 1$, then,

$$(r_{72}r_{80} - r_{70}r_{82})^2 + (r_{70}r_{81} - r_{71}r_{80})^2 - (r_{71}r_{82} - r_{72}r_{81})^2 = 0 \tag{60}$$

Rearranged (60) to a polynomial equation with the fourth degree. The coefficients L, M, N, P, and $Q$ are terms of $t_3$, as in Equation (61).

$$Lt_1^4 + Mt_1^3 + Nt_1^2 + Pt_1 + Q = 0 \tag{61}$$

Next, by substituting the trigonometric identities formula $S\varnothing_i = \frac{2t_{i}}{1+t_i^2}$ and $C\varnothing_i = \frac{1-t_i^2}{1+t_i^2}$ where $t_i = tan\frac{\varnothing_i}{2}$ into (42), the rearrangement of a second-degree polynomial equation can be obtained as (57). The coefficients $G$, $H$, and $I$ are also in terms of $t_3$

$$Gt_1^2 + Ht_1 + I = 0 \tag{62}$$

By substituting $t_1^2 = -\frac{I+Ht_1}{G}$ from (62) to (61), then, $t_1$ can be obtained as:

$$t_1 = \frac{LIH^2 - IG(LI + MH) + NIG^2 - QG^3}{-LH^3 + HG(2LI + MH) - G^2(MI + NH) + PG^3} \tag{63}$$

When substituting $t_1$ from (63) into (49), where $G0$; then,

$$
\begin{aligned}
G^3 \{ Q^2 G^4 - QPHG^3 \quad &-2QNIG^3 + IP^2 G^3 + QNH^2 G^2 + 3QMIHG^2 - PNIHG^2 \\
&-2PMI^2 G^2 + N^2 I^2 G^2 - QMH^3 G - 4QLIH^2 G + PMIH^2 G \\
&+3PLHI^2 G - NMHI^2 G - 2NLI^3 G + M^2 I^3 G + QLH^4 - PLIH^3 \\
&+NLI^2 H^2 - MLHI^3 + L^2 I^4 \} = 0
\end{aligned}
\tag{64}
$$

Subsequently, a 16th-degree polynomial equation was found from (65), in terms of $t_3$:

$$
Q^2 G^4 - QPHG^3 - 2QNIG^3 + IP^2 G^3 + QNH^2 G^2 + 3QMIHG^2 - PNIHG^2
$$
$$
-2PMI^2 G^2 + N^2 I^2 G^2 - QMH^3 G - 4QLIH^2 G + PMIH^2 G + 3PLHI^2 G-
\tag{65}
$$
$$
NMHI^2 G - 2NLI^3 G + M^2 I^3 G + QLH^4 - PLIH^3 + NLI^2 H^2 - MLHI^3 + L^2 I^4 = 0
$$

$t_3$ can be obtained by solving the 16th-degree polynomial Equation (65) using MATLAB and $t_1$ can be obtained by substituting $t_3$ in (63). $\varnothing_1$ and $\varnothing_3$ can be solved using $t_i = tan\frac{\varnothing_i}{2}$ where $\varnothing_i = 2tan^{-1}(t_i)$ then $\varnothing_3$ can be obtained from (59). After finding $\varnothing_1$, $\varnothing_2$ and $\varnothing_3$ and substituting in (34)–(36), then, $B_1$, $B_2$ and $B_3$ can be solved with knowledge of $\theta_7$, $\theta_8$ and $\theta_9$.

The position vector $O_7 = \begin{bmatrix} O_{7,x} & O_{7,y} & O_{7,z} \end{bmatrix}^T$, can be calculated by using Equation (66):

$$
O_7 = \frac{1}{3}[B_1 + B_2 + B_3]
\tag{66}
$$

Then computing orientation $R_B^A$ elements $u_x$, $u_y$ *and* $u_z$ By using (23)–(25) as (67)–(69).

$$
u_x = \frac{B_{1,x} - O_{7,x}}{d}
\tag{67}
$$

$$
u_y = \frac{B_{1,y} - O_{7,y}}{d}
\tag{68}
$$

$$
u_z = \frac{B_{1,z} - O_{7,z}}{d}
\tag{69}
$$

After calculating $u_x$, $u_y$ and $u_z$ the elements $v_x$, $v_y$ and $v_z$, can be obtained using (23)–(25).

$$
v_x = \frac{2(B_{2,x} - O_{7,x} + \frac{1}{2}du_x)}{\sqrt{3}d}
\tag{70}
$$

$$
v_y = \frac{-2(\sqrt{3}B_{2,x} - O_{7,y} + \frac{1}{2}du_y)}{\sqrt{3}d}
\tag{71}
$$

$$
v_z = \frac{2(B_{2,z} - O_{7,z} + \frac{1}{2}du_z)}{\sqrt{3}d}
\tag{72}
$$

Finally, by solving the orthogonal Equations (13)–(15), then, $w_x$, $w_y$ *and* $w_z$ can be obtained.

## 4. Modeling and Simulation

This section presents the proposed hybrid manipulator's numerical and graphical modeling.

### 4.1. Numerical Model

The kinematic modeling for the hybrid manipulator was written and applied using MATLAB software. The model was written as a script using the following parameters data $H = 455$ mm, $I = 35$ mm, $M = 400$ mm, $L = 99$ mm, $G = 420$ mm, $N = 25$ mm, $d = 56$ mm, $k = 56$ mm, $l_1 = 25$ mm, and $l_2 = 61$ mm. The hybrid manipulator kinematic model has the input represented in ($\theta_i$) and the outputs represented in position ($XYZ$) and orientation ($ABC$).

### 4.2. Graphical Model

The new hybrid manipulator was drawn with actual dimensions and assembled using SOLIDWORKS software. Then, it was imported into MATLAB to test the model using SIMSCAPE. A Monte Carlo algorithm was created and written to generate a cyclic input for ($\theta_i$) and record outputs ($XYZABC$) for the end effector of the hybrid manipulator. Two hundred thousand iterations were triggered, followed by filtering the data by removing the unwanted data due to the collision between the 3-RRS mechanism and the KUKA robot. In each iteration for forward kinematics, the program records the position and orientation of the serial-parallel hybrid manipulator end effector to generate a new workspace. This workspace includes thousands of points describing the end effector's translation and orientation. Furthermore, this workspace describes the reachability of the hybrid manipulator, as illustrated in Figure 6. The workspace for the KUKA robot arm with (+) sign, while the workspace for a hybrid manipulator with sign (×). Moreover, the final results from the numerical and graphical models were compared, and the error did not exceed more than 1% [32].

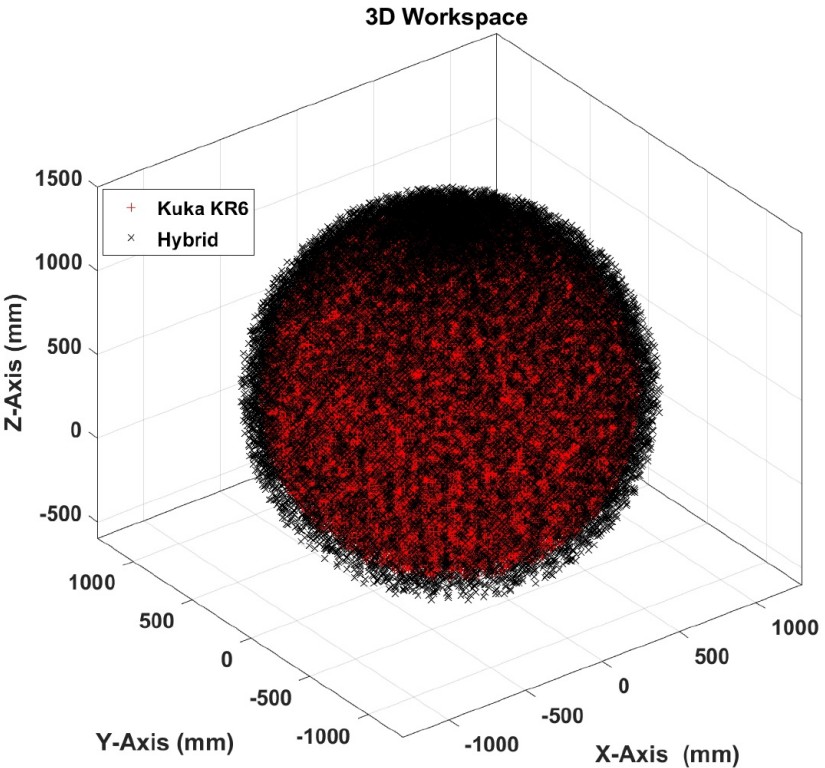

**Figure 6.** 3D workspace for KUKA KR6 R900 with (+) sign and hybrid manipulator with (×) sign.

### 5. Workspace Analysis

The 2D workspace layer was extracted from the 3D workspace, as shown in Figures 7a and 8a. The 2D workspace (XZ-plane) at X = 0 and (YZ-plane) at Y = 0 are presented, respectively, in Figures 7b and 8b, while the difference between the KUKA serial manipulator, with sign (+) in red, and the hybrid manipulator, with sign (×) with black, can be noticed in the inner and outer workspaces. However, Figure 7c shows that the hybrid manipulator workspace area covered 132.61% compared to the KUKA manipulator. Figure 8c shows that the hybrid manipulator workspace covered 128.33% compared to the KUKA manipulator. Furthermore, Figures 7d and 8d show the range of motion for the hybrid manipulator compared to the KUKA manipulator. In addition, Tables 3 and 4 present the dimension of the 2D workspace range at X = 0 and Y = 0, respectively.

**Table 3.** 2D Workspace Range at X = 0 (Side).

|  | X_Min (mm) | X_Max (mm) | Z_Min (mm) | Z_Max (mm) |
|---|---|---|---|---|
| **KUKA** | −931.2 | 981.2 | −422.9 | 1356 |
| **HYBRID** | −1044 | 1094 | −536 | 1469 |

**Table 4.** 2D Workspace Range at Y = 0 (Front).

|  | Y_Min (mm) | Y_Max (mm) | Z_Min (mm) | Z_Max (mm) |
|---|---|---|---|---|
| **KUKA** | −981.2 | 981.2 | −422.9 | 1356 |
| **HYBRID** | −1094 | 1094 | −536 | 1469 |

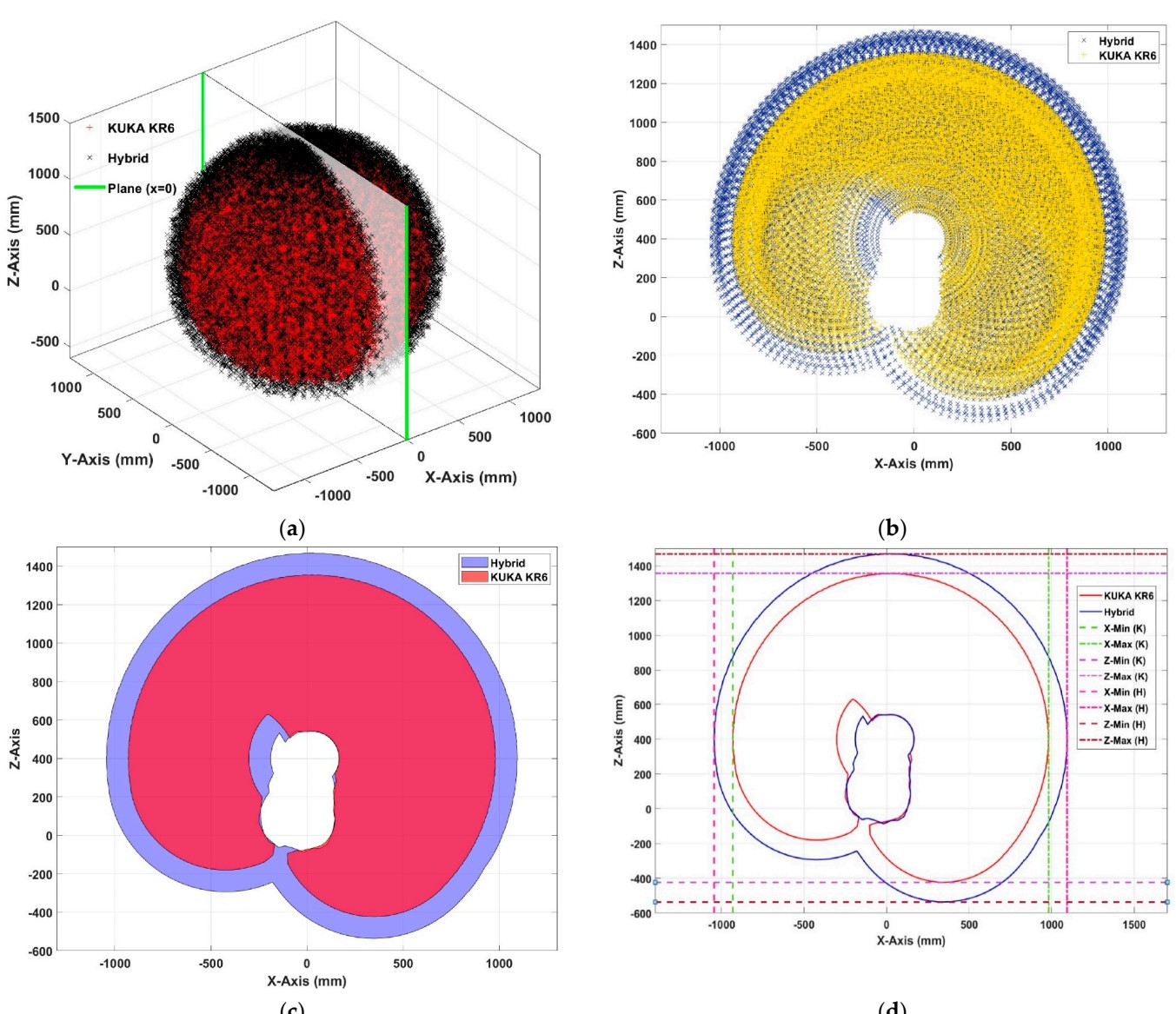

**Figure 7.** Workspace analysis at X = 0 : (**a**) extract slice from the 3D workspace at X = 0; (**b**) 2D workspace; (**c**) 2D workspace area; (**d**) limitation of the 2D workspace range.

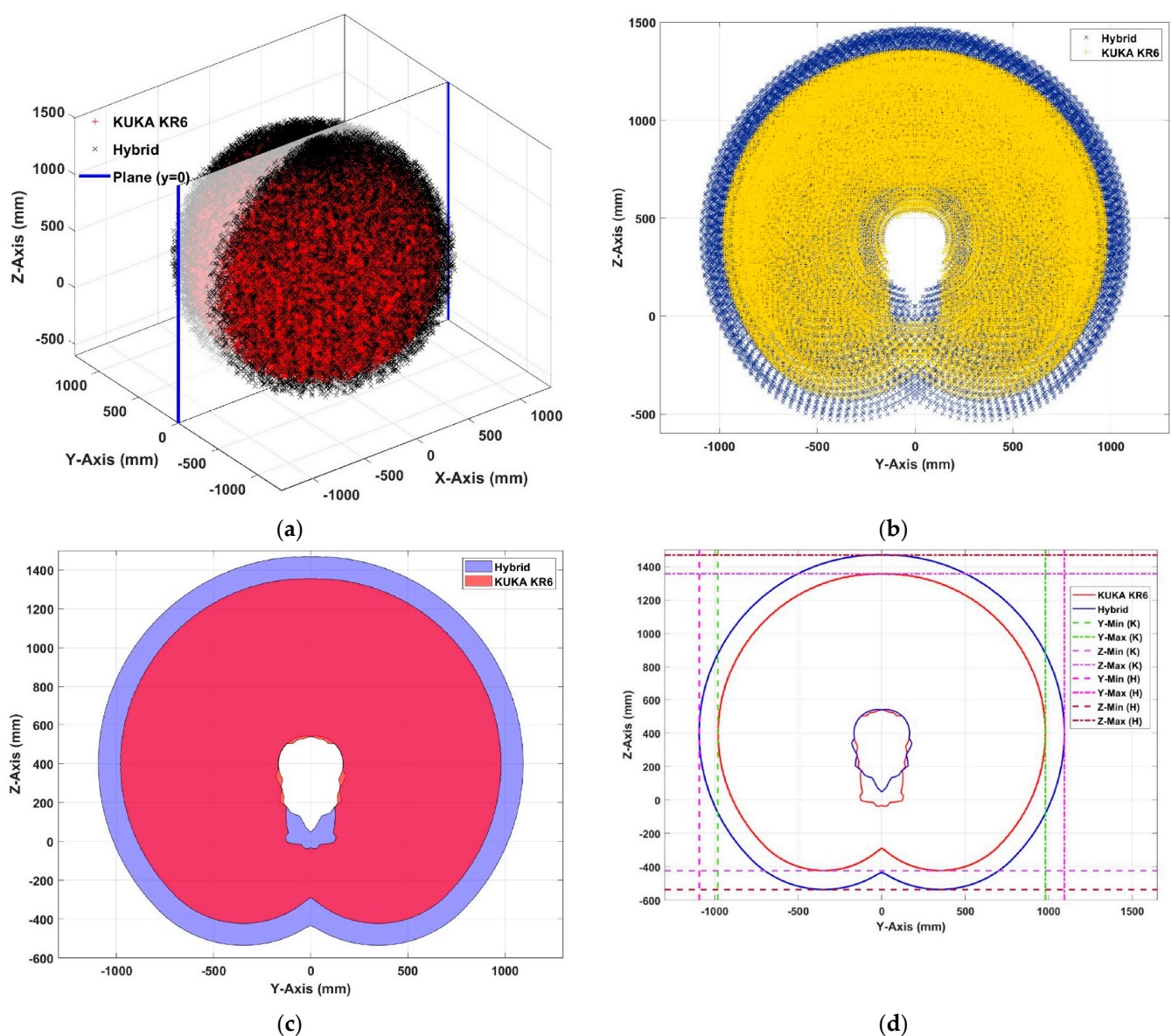

**Figure 8.** Workspace analysis at Y = 0 : (**a**) extract slice from the 3D workspace at Y = 0; (**b**) 2D workspace; (**c**) 2D workspace area; (**d**) 2D workspace range limitation.

## 6. Path Planning

The inverse kinematic must be solved to create a path for the new hybrid manipulator. However, the new serial-parallel robot has nine unknown angles, and it is complex to solve these unknown parameters; thus, we will trace a new method to follow the path. This method will depend on the previously obtained workspace and the extraction of a slice at the Z = 0 plane, as shown in Figure 9a. Furthermore, this workspace contains thousands of points with each point recording nine angles for the position and orientation of the new hybrid manipulator end effector. Figure 9b shows the selected surface (XY plane) after being extracted from the 3D workspace, while Figure 10 shows the selected points (A, B, C, D, E, F, G, and H), which represent the octagon path. Moreover, Table 5 shows the selected points with all nine angles ($\theta_1 \ldots \theta_9$), position (x, y, z), and orientation (A, B, C).

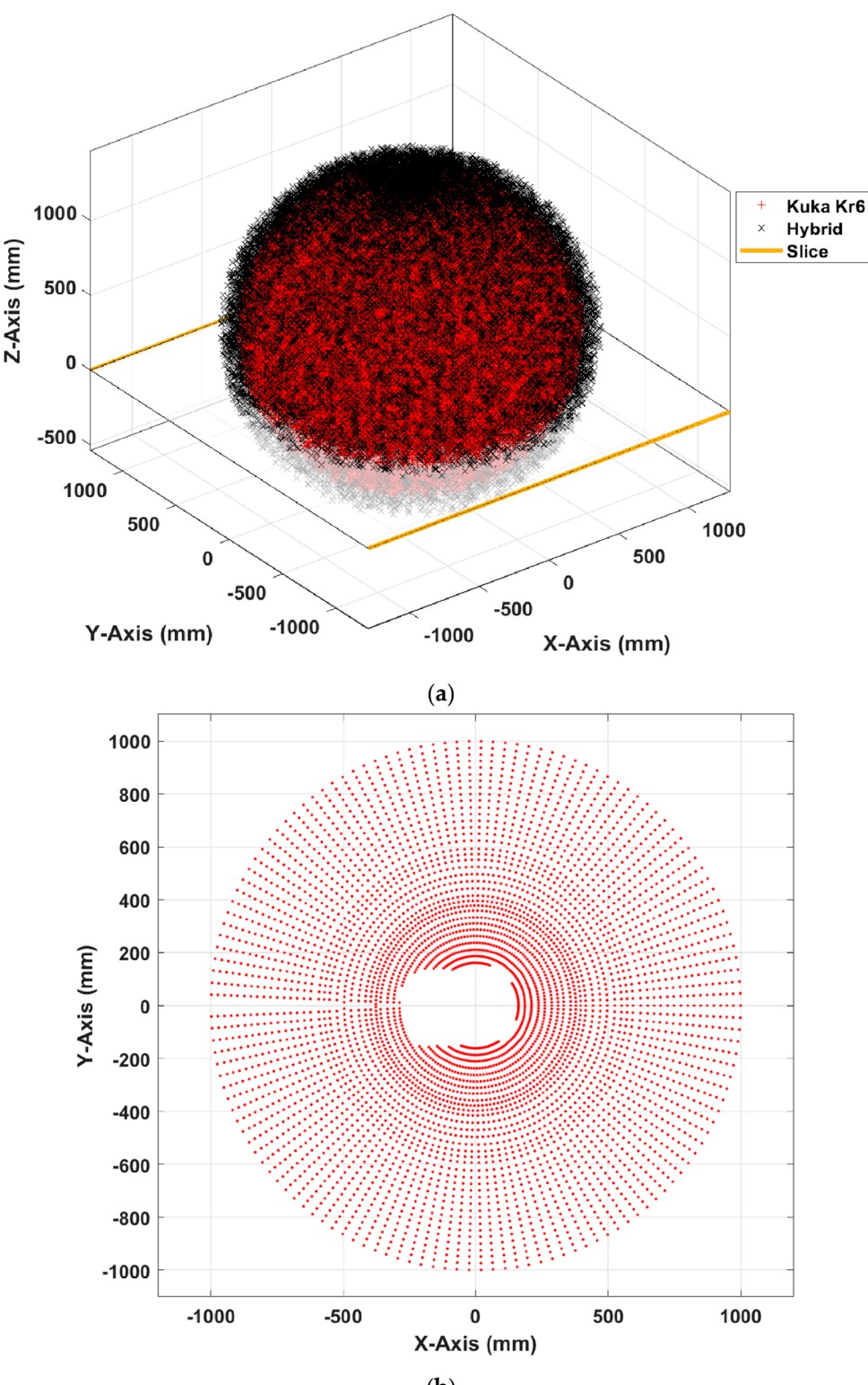

(**a**)

(**b**)

**Figure 9.** (**a**) Extract the slice from the 3D workspace at z = 0. (**b**) Top view of the 2D workspace at Z = 0.

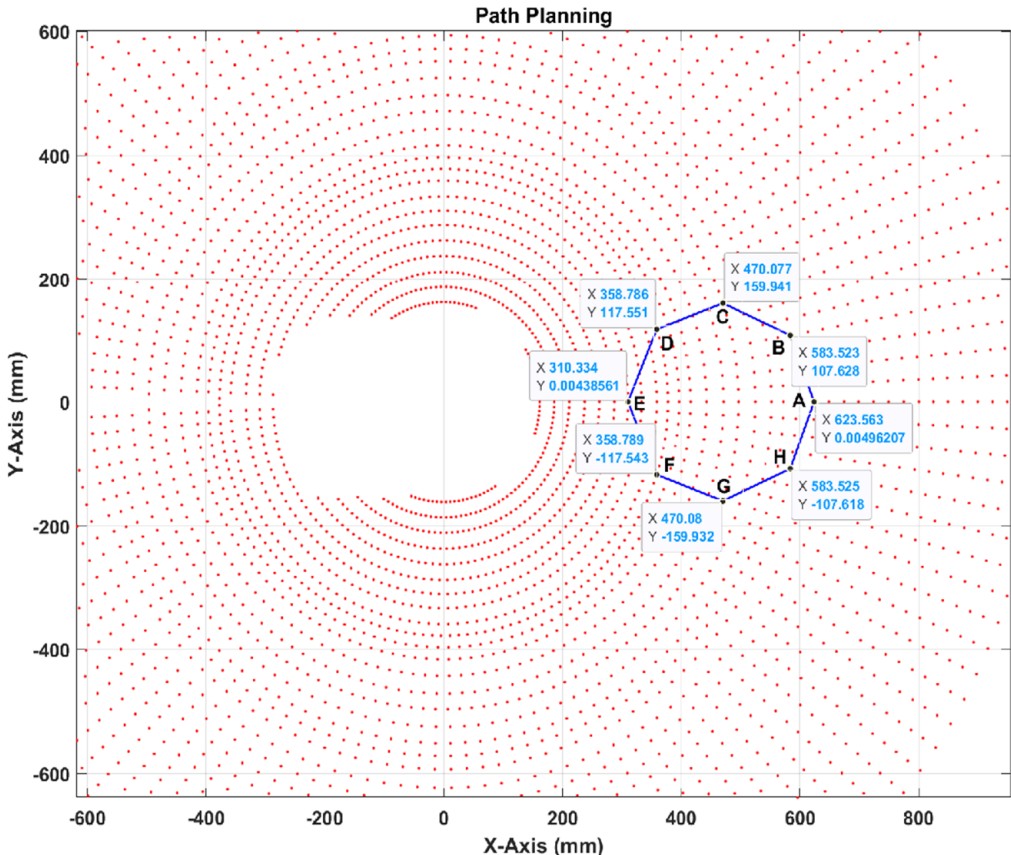

**Figure 10.** Octagon Path.

**Table 5.** Hybrid Manipulator Octagon Path Points (Simulation).

| Point | $\theta_1°$ | $\theta_2°$ | $\theta_3°$ | $\theta_4°$ | $\theta_5°$ | $\theta_6°$ | $\theta_7°$ | $\theta_8°$ | $\theta_9°$ | $x$ mm | $y$ mm | $z$ mm | $A°$ | $B°$ | $C°$ |
|---|---|---|---|---|---|---|---|---|---|---|---|---|---|---|---|
| A | 0.08 | −23.78 | 93.75 | 0 | 18.86 | −0.03 | 120 | 120 | 120 | 623.563 | 0.0049 | 0 | 179.88 | 1.17 | 180 |
| B | −10.23 | −26.26 | 98.28 | 0.02 | 19.15 | −0.03 | 90 | 90 | 90 | 583.523 | 107.628 | 0 | −169.78 | −1.17 | −180 |
| C | −18.62 | −32.10 | 114.85 | 0.02 | 8.59 | −0.03 | 60 | 60 | 60 | 470.077 | 159.941 | 0 | −161.39 | −1.34 | −180 |
| D | −17.94 | −34.85 | 133.23 | 0.03 | −9.15 | −0.03 | 30 | 30 | 30 | 358.786 | 117.551 | 0 | −162.06 | 0.77 | 180 |
| E | 0.08 | −33.47 | 143.07 | 0.03 | −22.52 | −0.03 | 0 | 0 | 0 | 310.334 | 0.0043 | 0 | 179.41 | 2.92 | 180 |
| F | 17.94 | −34.85 | 133.23 | 0.03 | −9.15 | −0.03 | 30 | 30 | 30 | 358.789 | −117.54 | 0 | 162.42 | 0.99 | 180 |
| G | 18.62 | −32.10 | 114.85 | 0.02 | 8.59 | −0.03 | 60 | 60 | 60 | 470.08 | −159.93 | 0 | 161.66 | −1.08 | −180 |
| H | 10.23 | −26.26 | 98.28 | 0 | 19.15 | −0.03 | 90 | 90 | 90 | 583.525 | −107.61 | 0 | 168.87 | 0.73 | −180 |

## 7. Experimental Setup and Testing

In the beginning, the 3-RRS parallel manipulator must be prepared to be attached to the KUKA KR6 R900 serial robot. It was designed using SOLIDWORKS software and implemented using a 3D printer. Moreover, it was provided with a thrust ball bearing at the passive joint ($C_i$) to prevent friction between the active link and the passive link. Furthermore, it will support the axial load on the joint, while at the spherical joint ($B_i$) a ball joint with a neodymium N-35 magnet was used for flexibility of movement, easy installation, and fast replacement with other end effectors, as shown in Figure 11. Finally, the 3-RRS was supported with three DYNAMIXEL servomotors (XL-430), which provide many features, such as serial communication, PID control, feedback of position angle, speed, temperature, and load.

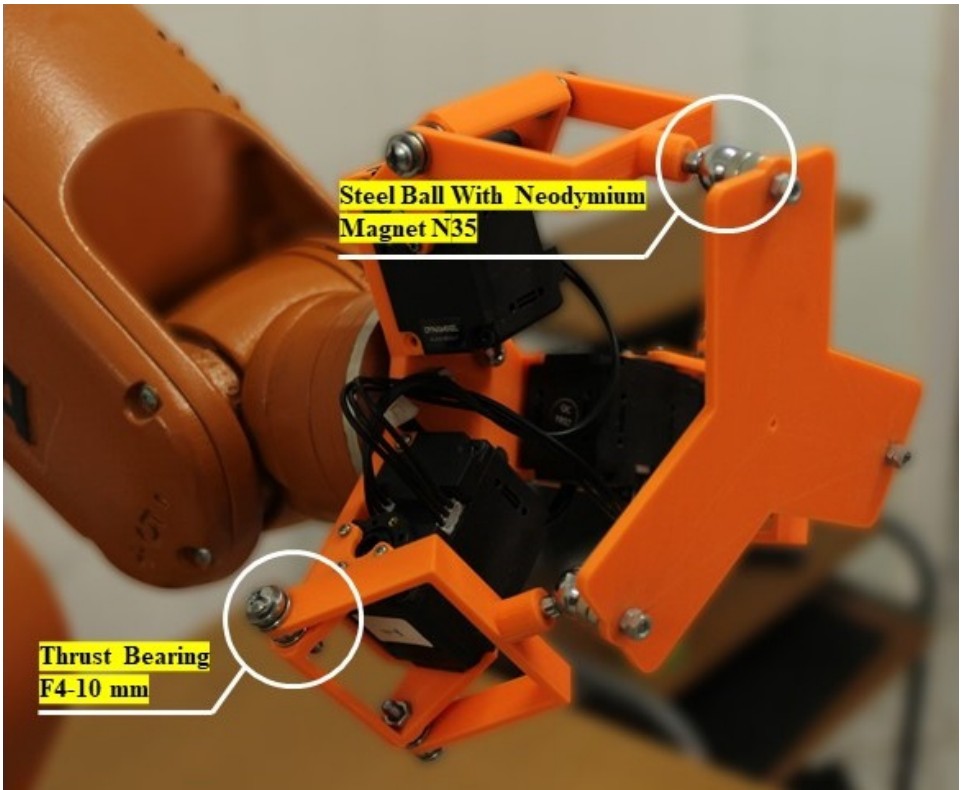

**Figure 11.** KUKA Kr6 R900 manipulator installed with 3-RRS.

Figure 12 shows the experimental setup, which contains the 3-RRS parallel robot mounted in the end effector of the serial manipulator KUKA KR6 R900. Then, the KRC4 KUKA controller was prepared and installed with KUKAVARPROXY, which can read or write variable data on the KUKA manipulator [33]. Further, it converts the controller to a server that can host up to ten clients. The KRC4 controller is connected via TCP/IP with the Raspberry Pi (client computer), which is provided with a Python code that can send the required joints to the KUKA controller and Arduino, via USB connection. Furthermore, Figure 13 describes the experimental process required to control the hybrid manipulator and follow the required path, where $\theta_{hr}$, $\theta_{sr}$, and $\theta_{pr}$ represent the required joint angles for the hybrid manipulator, the KUKA robot, and the 3-RRS parallel manipulator. Here, $\theta_{hf}$, $\theta_{sf}$, and $\theta_{pf}$ represent the joint angle feedback for the hybrid manipulator, the serial KUKA serial robot, and the 3-RRS parallel manipulator, respectively. The Python program includes the KUKA messenger protocol to send data (read/write) to the KUKA KRC4 controller through KUKAVARPROXY, while it has a 3-RRS messenger, which is responsible for communicating with the 3-RRS parallel manipulator through ARDUINO. Additionally, the KRL (KUKA Robot Language) program can actuate the KUKA robot from the data received via the KUKAVARPROXY. Finally, there is a digital input/output interface between the KUKA controller (KRC4) and the 3-RRS controller (Arduino) for synchronization between the KUKA manipulator and the 3-RRS manipulator.

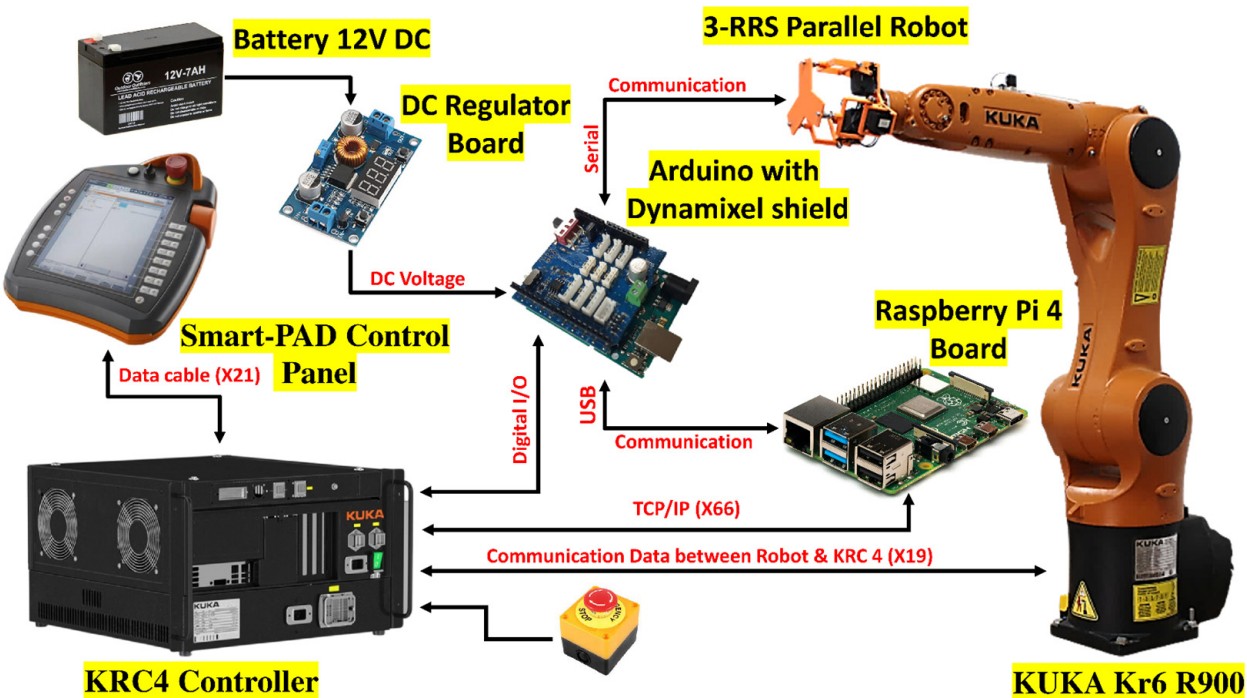

**Figure 12.** Experimental Setup Diagram.

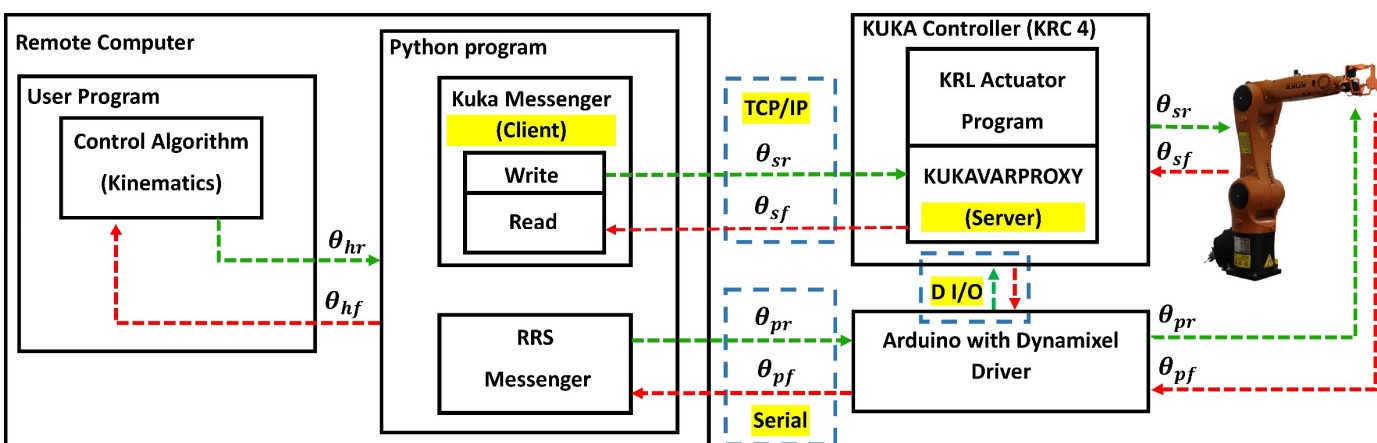

**Figure 13.** Experimental Setup Chart.

Figure 14 shows the actual laboratory experimental test that follows the octagon path following the specified eight points (A, B, C, D, E, F, G, and H), mentioned above from the workspace. Then, by recording the position $(x, y, z)$ and orientation of the points (A, B, C, D, E, F, G, and H), as shown in Table 6, and the angle of the serial-parallel manipulator joints, as shown in Figure 15, we found that the new SPHM follows the selected path precisely and correctly (https://clipchamp.com/watch/76B0d4mMP69 (accessed on 10 Septmber 2022)). The data were matched with the previous data recorded from the graphical model.

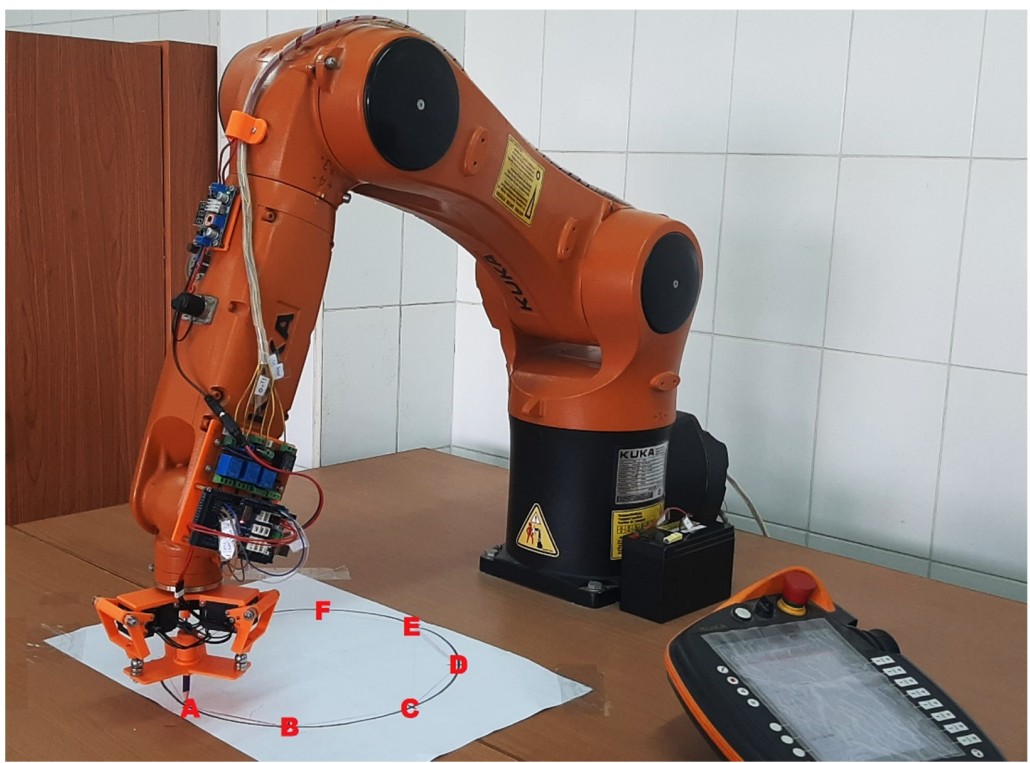

**Figure 14.** Experimental Setup of the Serial-Parallel Manipulator.

**Table 6.** Hybrid Manipulator Octagon Path Points (Experimental Setup).

| Point | $x$ (mm) | $y$ (mm) | $z$ (mm) | $A°$ | $B°$ | $C°$ |
|---|---|---|---|---|---|---|
| **A** | 623.563 | 0.0049 | 0 | 179.88 | 1.17 | 180 |
| **B** | 583.523 | 107.628 | 0 | −169.78 | −1.17 | −180 |
| **C** | 470.077 | 159.941 | 0 | −161.39 | −1.34 | −180 |
| **D** | 358.786 | 117.551 | 0 | −162.06 | 0.77 | 180 |
| **E** | 310.334 | 0.0043 | 0 | 179.41 | 2.92 | 180 |
| **F** | 358.789 | −117.54 | 0 | 162.42 | 0.99 | 180 |
| **G** | 470.08 | −159.93 | 0 | 161.66 | −1.08 | −180 |
| **H** | 583.525 | −107.61 | 0 | 168.87 | 0.73 | −180 |

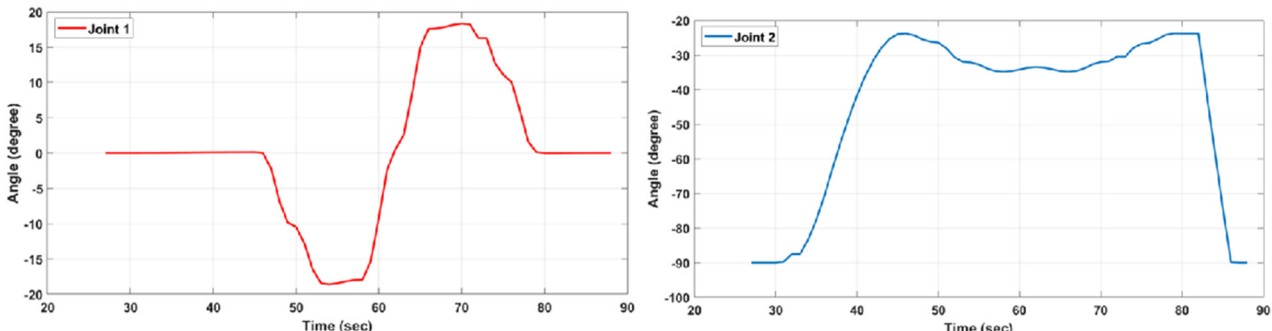

**Figure 15.** *Cont.*

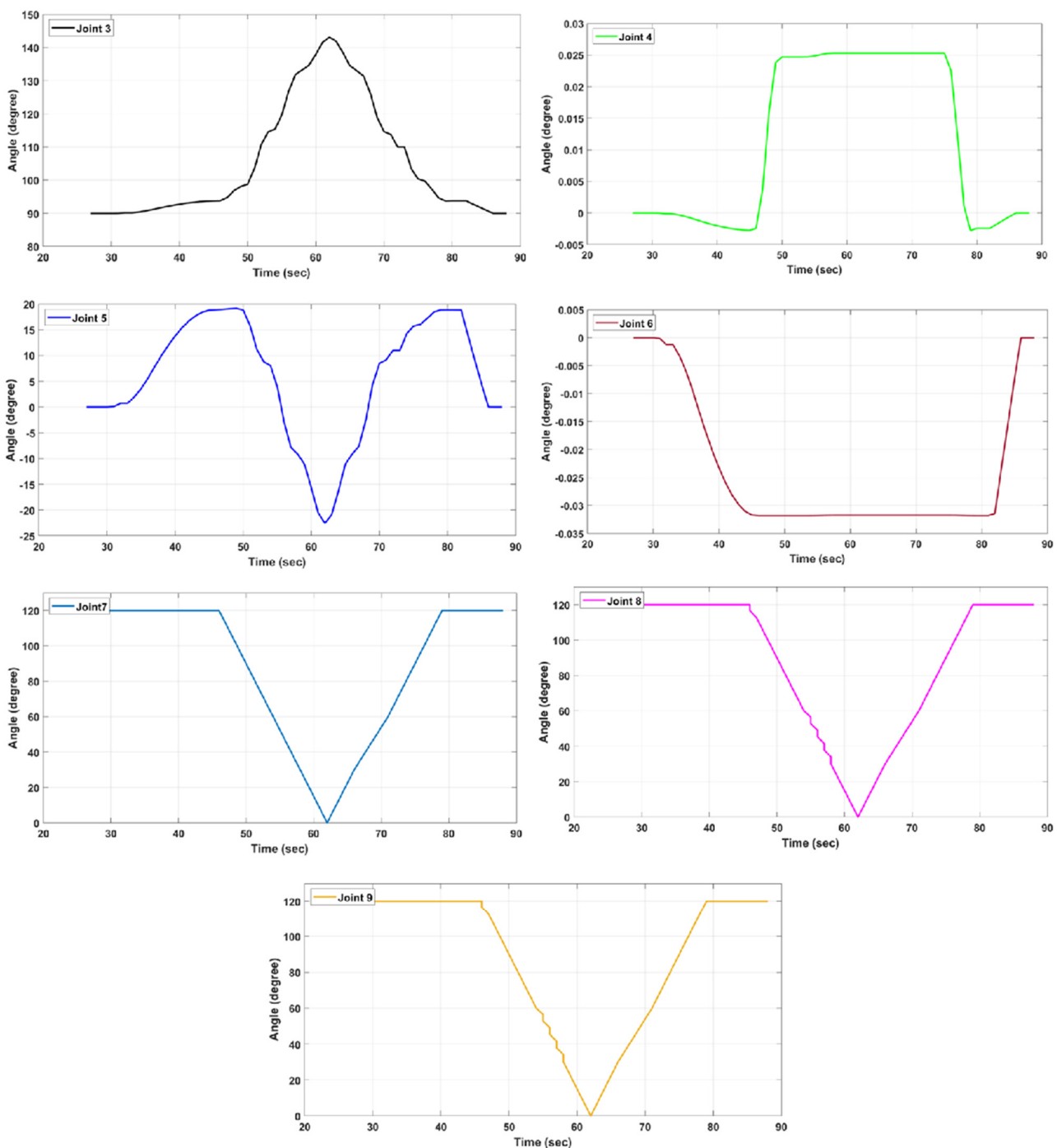

**Figure 15.** Serial-Parallel Manipulator Joints Angle.

## 8. Conclusions

A completely new serial-parallel hybrid manipulator robot is proposed to enlarge the workspace of the KUKA KR6 R900 serial manipulator. The new layout is to couple a 6-DOF KUKA KR6 R900 serial robotic arm with a 3-DOF 3-RRS parallel robot. This novel concept has the positioning and orientation potential to enhance the KUKA KR6 R900 end effector. A 3D CAD design was submitted for the suggested system, and a kinematic diagram for the new hybrid robot arm was delivered. Moreover, the kinematic solution for the new hybrid manipulator was obtained. Then, a graphical model with complete workspace analysis for the new SPHM was presented and discussed. The analysis declared that the hybrid manipulator workspace area was covered by 132.61 compared to the KUKA manipulator

in the XZ plane. In comparison, the percentage was 128.33 in the YZ plane. Finally, the results of the experimental laboratory test and the simulation were compared with those following an octagon path, and it was noticed that they matched.

**Author Contributions:** Conceptualization, methodology, and validation, M.E., E.M.F, S.A.R. and K.G.; formal analysis, M.E. and E.M.F.; investigation, M.E. and E.M.F.; resources, Y.H.H. and S.A.M.; data curation, M.E. and E.M.F.; writing—original draft preparation, M.E. and E.M.F.; writing—review and editing, H.Y. and K.G.; supervision, M.E., S.A.R. and S.A.M. All authors have read and agreed to the published version of the manuscript.

**Funding:** This research was funded by the Natural Science Foundation of Shandong Province (Grant no. ZR2022QH214). And The APC was funded by [ZR2022QH214].

**Institutional Review Board Statement:** Not applicable.

**Informed Consent Statement:** Not applicable.

**Data Availability Statement:** Not applicable.

**Conflicts of Interest:** The authors declare that they have no conflict of interest.

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
