# Peer review of "Workspace Analysis and Path Planning of a Novel Robot Configuration with a 9-DOF Serial-Parallel Hybrid Manipulator (SPHM)"

_applsci, doi:10.3390/app13042088_

Round 1
Reviewer 1 Report
1. I don’t see any logic in the Introduction section. The authors just list some of the existing works about hybrid manipulators. But what are the relationships between these works and the proposed one? Is there any research gap that needs to be filled? What are the contributions of this paper?
2. About the structure design, the authors may provide some discussions about why they choose the proposed structure (6-DOF serial robot +3-RSS parallel robot) since there are various combinations that can be adopted for hybrid manipulators. Is this structure designed for some specific applications?
3. The workspace of robots can be classified as translation workspace, orientation workspace, reachable workspace, dextrous workspace, and so on. What is the workspace type discussed in Section 5?
4. Is there any singular configuration inside the workspace? The authors may consider analyzing the singular conditions of the hybrid robot.
5. Int Section 6, I don't understand how the authors generate trajectories from the points in Table 5. The authors may provide more information about trajectory planning.
6. In Section 7, the authors just post the joint angles of the hybrid manipulator in Figure 16, without any analysis of the results. How is the trajectory following error? According to Figure 13 and Figure 14, it is noticed that the serial manipulator and the parallel manipulator appear to be controlled separately by the Arduino and KRC4. So the synchronized performance of the hybrid manipulator may be a problem, especially in performing dynamic trajectories.
7. In the video mentioned in Section 7, I see the robot reach the specified points (A, B, C, …) in sequence with zero speed (the robot is exactly stopped at each point). But in Figure 16, the presented joint trajectories are all continuous, which is not matched with the video. So can the authors explain this point?
Author Response
Point 1: I don’t see any logic in the Introduction section. The authors just list some of the existing works about hybrid manipulators. But
- what are the relationships between these works and the proposed one? Response: Thank you for your guidance. The proposed robot is an entirely new robot that handles the interaction between serial and parallel manipulators. The new robot is improving the overall workspace and developing the robot's orientation.
- Is there any research gap that needs to be filled?Response: Thank you for your guidance. Regarding the limited orientation in the serial manipulators, the newly developed robot is designed and improved to try to increase the overall orientation. Moreover, it has been presented that the new robot's overall workspace has been increasing with the increase of the orientation, and in some cases, the robot can handle objects near the robot body. This will lead to improving the area of industrial robots that can handle objects near the dead zone or behind the robot joint limits.
- What are the contributions of this paper?Response: The main contribution is to improve the industrial serial robot functionality by adding parallel robot
Point 2: About the structure design, the authors may provide some discussions about why they choose the proposed structure (6-DOF serial robot +3-RSS parallel robot) since there are various combinations that can be adopted for hybrid manipulators. Is this structure designed for some specific applications?
Response: the proposed model has been selected based on different criteria, first the large orientations that could be derived from this model type, second the light weight of the new model that will not add any extra payload over the serial manipulator. Third, a simple structure and even an efficient model could represent the hybrid serial parallel manipulators.
Point 3: The workspace of robots can be classified as translation workspace, orientation workspace, reachable workspace, dextrous workspace, and so on. What is the workspace type discussed in Section 5?
Response: This workspace includes thousands of points that describe the translation and orientation for the end-effector also this workspace describes the reachability of the hybrid manipulator
Point 4: Is there any singular configuration inside the workspace? The authors may consider analysing the singular conditions of the hybrid robot.
Response: In reality, we are preparing a singularity analysis. This is a new analysis we will present in a new article discussing the hybrid serial-parallel robot singularities.
Point 5: Int Section 6, I don't understand how the authors generate trajectories from the points in Table 5. The authors may provide more information about trajectory planning.
Response: In fact, after extracting the workspace layer at z=0, the trajectory path was selected to create a path for the hybrid manipulator
Point 6:
In Section 7, the authors just post the joint angles of the hybrid manipulator in Figure 16, without any analysis of the results.
Response: the graph for joint angles in figure 16 was collected from KUKA and 3-RRS real movement, and it was presented to compare with table 5, which was extracted from the workspace. It will be noticed that the joint angle in both was the same value.
According to Figure 13 and Figure 14, it is noticed that the serial manipulator and the parallel manipulator appear to be controlled separately by the Arduino and KRC4. So, the synchronized performance of the hybrid manipulator may be a problem, especially in performing dynamic trajectories.
Response: there is a digital input/output interface between the Arduino and the KUKA controller (KRC4) for synchronization between the KUKA manipulator and the 3-RRS manipulator.
Point 7: In the video mentioned in Section 7, I see the robot reach the specified points (A, B, C, …) in sequence with zero speed (the robot is exactly stopped at each point). But in Figure 16, the presented joint trajectories are all continuous, which is not matched with the video. So, can the authors explain this point?
Response: yes, we mean that to present the efficiency of the hybrid robot movement clearly

Reviewer 2 Report
I suggest to delete the words "robot configuration with" in the title.
The literature review should be improved adding at least these important references on hybrid manipulators:
- Zeng Q. et al Structural synthesis and analysis of serial-parallel hybrid mechanisms with spatial multi-loop kinematic chains (2012) Mechanism and Machine Theory, 49, pp. 198 - 215
- Zheng X.Z. et al Kinematic analysis of a hybrid serial-parallel manipulator (2004) International Journal of Advanced Manufacturing Technology, 23 (11-12), pp. 925 - 930
- Zeng Q., et al Design of parallel hybrid-loop manipulators with kinematotropic property and deployability (2014) Mechanism and Machine Theory, 71, pp. 1 - 26
- Romdhane L. Design and analysis of a hybrid serial-parallel manipulator (1999) Mechanism and Machine Theory, 34 (7), pp. 1037 - 1055
Moreover, at the end of the introduction it is important to mention that optimal motion planning methods for redundant manipulators such as the ones presented in these papers (cite at least these)
- Tringali A. et al Finite-horizon kinetic energy optimization of a redundant space manipulator (2021) Applied Sciences (Switzerland), 11 (5)
- Tringali A., et al Globally optimal inverse kinematics method for a redundant robot manipulator with linear and nonlinear constraints (2020) Robotics, 9 (3), art. no. 61
can also be applied to hybrid redundant manipulators (such as the one of this paper).
The workspace graphs are not clear, please find a way to make more easy to understand.
Author Response
Point 1: The literature review should be improved adding at least these important references on hybrid manipulators:
- Zeng Q. et al Structural synthesis and analysis of serial-parallel hybrid mechanisms with spatial multi-loop kinematic chains (2012) Mechanism and Machine Theory, 49, pp. 198 - 215
- Zheng X.Z. et al Kinematic analysis of a hybrid serial-parallel manipulator (2004) International Journal of Advanced Manufacturing Technology, 23 (11-12), pp. 925 - 930
- Zeng Q., et al Design of parallel hybrid-loop manipulators with kinematotropic property and deployability (2014) Mechanism and Machine Theory, 71, pp. 1 - 26
- Romdhane L. Design and analysis of a hybrid serial-parallel manipulator (1999) Mechanism and Machine Theory, 34 (7), pp. 1037 - 1055
Point 2: Moreover, at the end of the introduction it is important to mention that optimal motion planning methods for redundant manipulators such as the ones presented in these papers (cite at least these)
- Tringali A. et al Finite-horizon kinetic energy optimization of a redundant space manipulator (2021) Applied Sciences (Switzerland), 11 (5)
- Tringali A., et al Globally optimal inverse kinematics method for a redundant robot manipulator with linear and nonlinear constraints (2020) Robotics, 9 (3), art. no. 61
Response for point 1,2: Thank you for your guidance. Really these references have increased the value of the literature and we have already added all of them as it has been presented in the literature in numbers 4,5,6,7 and 25, 26
Point 3: The workspace graphs are not clear, please find a way to make easier to understand.
Response: we have added a description that show the illustration the workspace in section as follow “ and it can be noticed the difference between the KUKA serial manipulator with sign (+) in red and the hybrid manipulator with sign (x) with black in the inner and outer workspace “ a new graph has been presented to show the workspace in easier way.

Author Response
Point 1: The authors have to revise their manuscript linguistically and to improve the writing style. It contains some typos.
Response: Thank you for your guidance. the paper has been revised and changed totally in the language.
Point 2: The introduction section must be reorganized. The authors are first discussing the category of SP robots in page 2, the HPS ones are following at the bottom of the same page, and then they resume their discussions of PS robots in page 3.
Response: thank you for being appreciated effort we have reorganized it.
Point 3: “[…] actuation distribution of a novel 3-(3RR1S) mechanism.” Contains a confusing notation/acronym. It should be simply 3-RRS (avoid any redundancy 3-3 and unnecessary numbering 1S)
Response: all notations were explained and changed.
Point 4: What is the meaning of the acronym RCM?
Response: remote center motion (RCM) and it was already defined in the paper
Point 5: “3T2R” is used without any prior definition. T and R must be first used in full text and then referred to; such as translation (T) and rotation (R). For example, at the bottom of page 2, one can read “The proposed manipulator allows for the accomplishment of five distinct moves, three for translations (3T) and two for rotations (2R) movement patterns.”
Response: all definition was explained in the paper
Point 6: The authors are required to add/refer-to the movement orientation of each joint on the CAD models shown in figures 1 and 3. Alternatively, the authors would add in a separate figure a symbolic representation of the proposed robotic configuration including the moving joints w.r.t. their axes of rotation respectively.
Response: the required edit and figures were added
|
|
Point 7: The authors first intend to use the DH convention (mostly the classical one used in the paper). This is fine. They do not use the common algorithm to assign the axes; for example, it is very common to refer to attribute ? to the axis of motion (rotation of any joint in this case). This is not the case as shown in figure 5. Again, this is fine, but assuming/considering the set of frames shown in figure 5, the parameters established in table 2 (of DH parameters) do not match with their corresponding configurations in figure 5. Roughly, one must read +90°, −? in first row, −? and ?2 + 90° in 2nd row, etc. (other typos or computation errors in the same table a priori).
Response: the table was modified with the correct parameters
Point 8: From this point, computation errors raise in Eqs. (2) to (7). One would mention that even by assuming the table is correct, errors within these equations are noticed
Response: the equations (2) to (7) was edited
Point 9: Assuming Eqs. (8) to (28) are correct, there is a mistake in Eq. (31). One must read − 3??? 2 instead of + 3??? 2. This small typo unfortunately affects the following equations (32) and (33)
Response: we apologize, just mistake while typing equation
Point 10: Based on the frames and angles represented in figure 4.a, the terms of ?1, ?2, and ?3 in Eq. (34)-(36) are fully incorrect. The reasoning and calculation established in equations (37) to the end is highly impacted by these errors.
Response: the 3-RRS diagram was modified with correct angles

Reviewer 4 Report
The paper is very interesting paper which presents workspace analysis and path planning for a 9-DOF robot. The 9-DOF robot used in this experiment benefits from a serial 6-DOF industrial robot with a parallel manipulator. My comments are as follows.
Please give more information about the parallel manipulator with 3-RRS (three joints with Revolute-Revolute-Spherical) used in this paper and explain why is this manipulator chosen?
What is the benefit of having a 9DOF robot over a 6DOF one?
Figure 6 is not necessary and can be deleted.
Figure 8.b needs to be more clear.
Figure 14 needs to be more clear.
Author Response
Point 1:
- A) Please give more information about the parallel manipulator with 3-RRS (three joints with Revolute-Revolute-Spherical) used in this paper
- B) and explain why is this manipulator chosen?
Response 1(A):
The 3RRS (Three Revolute Revolute-Spherical) parallel robot is a type of parallel manipulator that consists of three revolute joints, arranged in a tripod configuration. It is known for its ability to handle payloads and maintain high precision in motion, making it a popular choice for applications in areas such as material handling, assembly, and machining. The 3RRS parallel robot features a spherical wrist that allows for a large workspace, as well as a compact design that allows for ease of installation in a variety of environments. Additionally, the three revolute joints provide high dexterity, allowing the robot to access hard-to-reach areas and perform complex tasks. The 3RRS parallel robot is controlled by actuators at each joint, which are coordinated through sophisticated control algorithms to achieve the desired motion. Overall, the 3RRS parallel robot is a versatile and reliable robotic solution for many industrial applications.
The 3-RRS design manufacturing is based on PLA+ material using a 3D printer. Each limb consists of a servo motor (active link) with a range motion of . The passive link with a U shape is connected with an active link using two thrust bearings (one for each side) with an inner diameter of 4 mm and outer diameter of 10 mm, as in figure 15(a) to prevent the friction between active link and passive link also to support the axial load on the joint. At the end of the limb, a spherical joint is represented in a steel ball with a diameter of 12 mm, and a flat round magnet as in figure 15(b) for more flexible movement. The magnet employed is neodymium N-35 which provides sufficient force to hold the moving plate. Also, it helps to reassemble the moving plate with the passive link if a collision happens. The 3-RRS parallel manipulator attached to the flange of the KUKA KR6 R900 serial manipulator as in figure 16.
Design Details:
The base was designed as an equilateral triangle shape with a central angle of 120ÌŠ see figure 17. Furthermore, there was a keenness on the size of the base to not impede the movement of the KUKA arm robot. Also, the 3-RRS base can hold three servo motors for the three limbs. Each servo motor is fixed at the base using four screws. The base can be set at the KUKA end joint with three (5 mm) screws.
The second part are limbs. In this design, there are three limbs, and each limb consists of three sections, the first section is a servo motor with an arm in a U shape (active link) and another U shape (passive link) fixed with an active link by two thrust bearing (one on each side), and at the end of the limb, there is a spherical joint. The servo motor used to drive the limb (active joint) is DYNAMIXEL XL430-W250 see figure 18, which provides full 360-degree rotation, high torque, small size, low weight, high resolution, position feedback, speed feedback, and load feedback. This servo motor translates the movement to the active link through the active joint, and figure 19 shows the active link dimensions.
Moreover, the passive link was designed as U-shape and ended with a steel ball with 12 mm diameter and coupled with a flat round magnet to act as a spherical joint to hold and allow the moving plateform to move in 3-DOF (roll, pitch, and heave) see figure 20.
Finally, the moving plate is very similar to the base, with an equilateral triangle shape and a central angle of 120ÌŠ. A three-flat round neodymium N35 magnet was fixed as in figure 21, which completes the spherical joint with the steel ball in the passive link.
Response 1(B): this type of manipulator was chosen because of its advantages and a three main reasons:
The 3RRS parallel robot has several advantages, including:
- High Payload Capability: The 3RRS parallel robot is capable of handling heavy payloads, making it suitable for tasks that require precision handling of heavy objects.
- Large Workspace: The spherical wrist of the 3RRS parallel robot allows for a large workspace, providing flexibility in performing tasks in various orientations and positions.
- Compact Design: The 3RRS parallel robot has a compact design, making it easy to install and operate in confined spaces.
- High Precision: The 3RRS parallel robot is known for its high precision in motion, ensuring accurate and consistent performance for demanding applications
- High Dexterity: The three revolute joints of the 3RRS parallel robot provide high dexterity, allowing the robot to access hard-to-reach areas and perform complex tasks.
- Reliability: The 3RRS parallel robot is a reliable solution for industrial applications, providing consistent and efficient performance.
- Versatility: The 3RRS parallel robot is versatile, making it suitable for a wide range of industrial applications, including material handling, assembly, and machining.
The first one: that the 3-RRS have only three degree of freedom which makes the mathematical model not complicated.
The second one: this manipulator provides one translation movement and two orientations instead of using parallel manipulator as delta robot which provides three translation movement with no orientation.
The third one: the combination of 3-RRS to the serial manipulator KUKA Kr6 will increase the overall robot orientation redundancy. The 9- DOF new robot will have at any position different orientation at the same point so this will improve the robot capability for grasping objects at any orientation with multiple solutions.
Point 2: What is the benefit of having a 9DOF robot over a 6DOF one?
Response: Regarding the limited orientation in the 6-DOF serial manipulators, the newly developed robot with 9-DOF is designed and improved to try to increase the overall orientation. Moreover, it has been presented that the new robot's overall workspace has been increasing with the increase of the orientation, and in some cases, the robot can handle objects near the robot body. This will lead to improving the area of industrial robots that can handle objects near the dead zone or behind the robot joint limits. Moreover, the following reasons are supporting our idea:
- Increased Range of Motion: With three additional degrees of freedom, a 9DOF robot can move in a wider range of directions, providing greater flexibility and versatility in performing tasks.
- Improved Complexity of Tasks: The additional degrees of freedom in a 9DOF robot allow it to perform more complex tasks, including tasks that require multi-axis movements or orientations.
- Enhanced Precision: The additional degrees of freedom in a 9DOF robot can lead to increased precision in motion, improving the accuracy of the robot's movements
- Better Adaptability: The additional degrees of freedom in a 9DOF robot allow it to better adapt to the requirements of a particular task, providing greater efficiency and effectiveness.
- Improved Agility: The additional degrees of freedom in a 9DOF robot can provide improved agility, allowing the robot to maneuver more quickly and easily to perform tasks.
- Greater Control: The additional degrees of freedom in a 9DOF robot provide greater control, allowing for more nuanced and sophisticated movements to be executed.
- Increased Robustness: The increased number of joints in a 9DOF robot can provide greater stability and robustness, improving the overall reliability and durability of the robot.
Point 3: Figure 6 is not necessary and can be deleted.
Response: Thank you for your guidance and the figure was deleted
Point 4: Figure 8.b needs to be clearer.
Response: a new figure was added after change the colours of data points
Point 5: Figure 14 needs to be more clear.
Response: a new figure was added after bold the words
Round 2
Reviewer 1 Report
Most of my comments are properly addressed, and I am satisfied with this revision. I think this paper can be accepted in the current version.
Author Response
Point 1: English language and style are fine/minor spell check required
Response: Thank you for your guidance. The paper has been revised, and all the minors in the spelling and style have been checked and corrected. Moreover, the new version has been attached to the reply.
